# Antioxidant, Anti-Inflammatory, and Antiproliferative Activity of a Callus Culture of *Prionosciadium dissectum* (Apiaceae)

**DOI:** 10.3390/plants14091394

**Published:** 2025-05-06

**Authors:** Antonio Bernabé-Antonio, Jessica Nayelli Sánchez-Carranza, José Antonio Silva-Guzmán, Antonio Romero-Estrada, Samantha Guadalupe Pérez-Rodríguez, Francisco Cruz-Sosa, Mariana Sánchez-Ramos, Aurelio Nieto-Trujillo

**Affiliations:** 1Department of Wood, Pulp and Paper, University Center of Exact Sciences and Engineering, University of Guadalajara, Km 15.5 Guadalajara-Nogales, Col. Las Agujas, Zapopan 45100, Jalisco, Mexico; jantonio.silva@academicos.udg.mx; 2Faculty of Pharmacy, Autonomous University of the State of Morelos, Av. Universidad 1001, Col. Chamilpa, Cuernavaca 62209, Morelos, Mexico; jessica.sanchez@uaem.mx; 3Technological Institute José Mario Molina Pasquel and Henríquez, Higher Technological Institute of Jalisco, Tala Academic Unit, Tecnológico Avenue S/N, Tala 45300, Jalisco, Mexico; are@uaem.mx; 4Department of Botany and Zoology, University Center of Biological and Agricultural Sciences, University of Guadalajara, Cam. Ramón Padilla Sánchez 2100, Col. Las Agujas, Zapopan 44600, Jalisco, Mexico; samantha.perez9153@alumnos.udg.mx; 5Department of Biotechnology, Autonomous Metropolitan University-Iztapalapa Campus, Av. Ferrocarril de San Rafael Atlixco 186, Col. Leyes de Reforma 1ª. Sección, Alcaldía Iztapalapa, Mexico City 09310, Mexico; cuhp@xanum.uam.mx (F.C.-S.); marianasan_06@xanum.uam.mx (M.S.-R.); 6Biotic Resources Research Center, Autonomous University of the State of Mexico, Carretera Toluca-Ixtlahuaca Km 14.5, Col. San Cayetano, Toluca 50295, State of Mexico, Mexico; anietot_ext@uaemex.mx

**Keywords:** Apiaceae, medicinal plant, plant growth regulator, in vitro culture, biological activity

## Abstract

Traditionally, medicinal plants have served as the main resource for treating various human health conditions. *Prionosciadium dissectum* is a plant used in traditional medicine in the southern region of Jalisco, Mexico, to treat inflammatory respiratory problems. However, this species has not undergone pharmacological or biotechnological studies that validate these popular uses. The aim of this study was to induce calluses on *P. dissectum* leaves and then evaluate the antioxidant, anti-inflammatory, and antiproliferative activity of their extracts. The best callus induction was obtained using Murashige and Skoog (MS) culture medium with 1 mg/L 2,4-dichlorophenoxyacetic acid (2,4-D) and 1 mg/L kinetin (KIN). Extracts of hexane, dichloromethane, and methanol were obtained from the dry biomass, and the highest yield was obtained with methanol. The total phenolic content and antioxidant activity of the methanolic extracts were quantified. The methanolic extract showed 26.5 ± 0.4 mg equivalents of gallic acid/g extract, while, for antioxidant activity, it demonstrated IC_50_ values of 49.4 ± 0.2 and 10.0 ± 0.0 μg/mL for 2,2-diphenyl-1-picrylhydrazyl (DPPH) and ((2,2′-azino-bis(3-ethylbenzothiazoline-6-sulfonic acid)) (ABTS), respectively. Regarding anti-inflammatory potential, the extracts did not significantly affect cell viability in RAW 264.7 macrophages. In contrast, it was clear that all extracts significantly decreased nitric oxide (NO) production at concentrations of 5–40 µg/mL. Additionally, extracts evaluated in human cancer cell lines only had a significant inhibitory effect at 100 µg/mL after 48 h, mainly with dichloromethane extract. This first biotechnological study indicates that *P. dissectum* cell cultures may produce compounds that favor the biological activities evaluated; however, it is necessary to carry out more in-depth evaluations of its extracts. This study is the basis for future research to enable the sustainable use of this valuable resource.

## 1. Introduction

Recently, the global surface area available for planting crops has been rapidly reduced due to the increase in the human population; it is, therefore, necessary to look for biotechnological alternatives, both to producing agricultural crops and to obtain value-added resources such as phytochemicals. Even the production of phytochemicals can be improved in a sustainable manner [1,2]. Several phytochemicals can help mitigate various diseases, among many other applications [3,4]. For instance, the prevalence of inflammatory diseases and cancer demands the continual search for new alternatives for their treatment because current drugs continue to present side effects; however, some anti-inflammatory agents have limited use in cancer therapy due to their toxicity and are required to be administered in cycles to reduce toxicity and prevent resistance [5,6]. In this case, it is necessary to search for new agents that can overcome these limitations. Plants represent one of the main sources of molecules for the development of new drugs, increasing the interest in the search for medicinal substances from plants to minimize tissue damage and provide greater comfort to the patient [7]. Among several plant families, the Apiaceae family is one of the most important flowering plants and includes several medicinal species that humanity has used since ancient times [8]. Ethnobotanical evidence reports the use of several of these species to alleviate various human ailments such as digestive issues, endocrine issues, reproductive issues, and respiratory system issues [9,10]. Regarding scientific studies, previous reports have revealed their importance as a resource with different biological activities, such as acaricidal [11], antibacterial [12,13], antiproliferative [14,15,16], anti-inflammatory [17,18,19,20], analgesic, radical scavenging [21], and antiviral activity [22]. Several of these properties are attributed to important compounds such as terpenoids, saponins, triterpenoids, coumarins, polyacetylenes, and steroids with biological activity [9,23], as well as polyphenols and essential oils [24].

*Prionosciadium dissectum* is one of the species growing in the mountainous regions of western Mexico [25]. The residents of the coast of the state of Jalisco use this plant to treat respiratory and skin ailments; however, there are no scientific reports that support these properties. There are some phytochemical studies for closely related species such as *Prionosciadium watsonii* from which extracts of the aerial part have been combined with CH_2_Cl_2_-MeOH (1:1), and, through the guided fractionation of a phytotoxic extract, some pyranocoumarins and pyranochromones have been isolated [26]. In another study carried out on *Prionosciadium thapsoides*, the dried roots were extracted using hexane, and the dihydrofurochromones (*S*)-(+)-4′-*O*-angeloyl-5-*O*-methylvisamminol and (*S*)-(+)-4′-*O*-senecioyl-5-*O*-ethylvisamminol and the coumarins jatamansin, buchtormin, isopterixin, isosamidine, psoralen, and bergapten were isolated [27]. However, the excessive exploitation of wild plants is not a viable option with regard to the environment. To our knowledge, no chemical or biotechnological studies have been carried out on *P. dissectum* that help with eco-friendly management to obtain secondary metabolites, using cell culture as a sustainable alternative while protecting wild plant populations. This study aimed to establish a callus culture from *Prionosciadium dissectum* leaves, obtain extracts, and evaluate the antioxidant and anti-inflammatory potential in RAW 264.7 macrophages and their antiproliferative potential in cancer cell lines.

## 2. Results and Discussion

### 2.1. Callus Culture

Leaf explants obtained from the potted plants (Figure 1a) were placed in MS culture medium to assess callus induction (Figure 1b). The PGR-free culture medium (control) and the medium containing only kinetin (KIN) did not result in any response in the explants; therefore, the explants turned brown and died after two weeks. In contrast, treatments containing 2,4-dichlorophenoxyacetic acid (2,4-D) alone or combined with KIN exhibited explant elongation, and the green explants turned a yellowish color. After two weeks of culture, all explants treated with 2,4-D formed calluses on the leaf veins or in the incision area, although some explant sections were dried (Figure 1c). Calluses containing 2,4-D with KIN showed rapid growth and were yellowish in appearance (Figure 1d).

The statistical analysis showed significant differences (*p* ≤ 0.05) between treatments (Figure 2). An increase in the percentage of callus induction was observed as the concentration of 2,4-D was increased until 100% induction was obtained with 1 mg/L 2,4-D plus 1 mg/L KIN. However, when using 1.5 mg/L 2,4-D, the percentage of callus induction decreased independently of the KIN concentration, which ranged from 62.5 to 68.8% (Figure 2). All calluses maintained their light yellowish and friable appearance; however, only calluses obtained using 1 mg/L 2,4-D plus 1 mg/L KIN were considered for frequent subcultures every 30 days since they showed adequate friability and growth.

To date, there are no reports of callus induction in the genus *Prionosciadium*, although there are studies on other genera of the Apiaceae family; for instance, in a study carried out on *Angelica archangelica*, it was found that 2,4-D combined with 6-Benzylaminopurine (BA) is the best substance for callus induction and maintenance treatment; however, the authors do not specify the concentrations of the best treatment [28]. A study on *Angelica sinensis* reported optimal conditions to induce calluses using 0.5 mg/L BA and 0.7 mg/L 2,4-D or 1.0 mg/L 2,4-D and 0.2 mg/L BA [29]. In another study on *Cnidium officinale*, callus induction was observed on root explants cultured with 0.5 mg/L 2,4-D and 0.5 mg/L BA [30]. In studies carried out on *Ferula ferulaeoides*, the induction of friable calluses followed the application of 2 mg/L indoleacetic acid (IAA) and 0.5 mg/L BA [31]. The use of 1 mg/L BA and 0.1 mg/L naphthaleneacetic acid (NAA) on explants of *Arracacia xanthorrhiza* apices has also been reported, which resulted in a higher number and increased growth of shoots [32]. In their recent study on *Ammi visnaga*, L. Golkar et al. [33] induced friable calluses using 3 mg/L BAP and 1 mg/L NAA. In *Foeniculum vulgare*, a high frequency of callus induction was obtained using 1.5 mg/mL BA and 0.5 mg/L NAA [34]. For other cultivated species, such as *Cuminum cyminum*, it was found that the best treatment for callus induction was 2.5 mg/L 2, 4-D, and 0.5 mg/L KIN [35]. The above reports indicate that the addition of a combination of auxin and cytokinin in a balanced cocktail induces cell division and promotes cell proliferation in cell culture [36]. The concentrations used in those studies are close to those used for *P. dissectum* in the current study, in which the combined application of 2,4-D and BA proved suitable for inducing friable callus growth, with the 1 mg/L concentration being particularly effective.

### 2.2. Callus Culture Growth

The growth kinetics of *P. dissectum* calluses (Figure 3) showed slow and constant growth until day 15; subsequently, a slight decrease was observed on day 20. Then, exponential growth was displayed until day 30, during which 7.6 g/L dry biomass was obtained as the maximum yield. The stationary phase was not observed since, after day 30, the callus had a sudden decrease in growth (death phase) until day 45, and the appearance of the callus turned brown. In a previous study on *Angelica sinensis*, it was reported that of the growth kinetic parameters of calluses cultured in four different media, Gambor (B5) and Schenk and Hildebrandt (HS), were superior to MS or White medium, using 2,4-D and BA [29]. In the callus culture of *Cnidium officinale*, the highest fresh weight of biomass was also reported after 30 days of culture when 2,4-D and BA were used as plant growth regulators [30].

In another species, *Jatropha curcas*, callus cultures showed a sigmoid growth, wherein the stationary phase reportedly lasted for 10 days, the exponential phase lasted up to 60 days, the line phase lasted between 60 and 100 days, and the death phase began at 130 to 180 days [37]. In *Costus speciosus* callus cultures, the maximum accumulation of dry biomass was obtained at week 5 and was allowed to grow until week 9 [38]. On the other hand, the leaf callus cultures of *Calophyllum brasiliense* had a 10-day adaptation phase and an exponential phase that lasted up to 40 days, obtaining a maximum of 18.72 g/L of dry biomass, after which the growth decreased [39]. The callus culture growth in the above studies was very slow compared to the *P. dissectum* cultures in the present study, in which the death phase began on day 30 (Figure 4).

### 2.3. Yield of Callus Extracts

The methanolic extract demonstrated a higher yield (17.9%) than the dichloromethane (1.0%) and hexane extracts (0.4%). This trend has also been observed for *Eryngium foetidum*, for which yields with methanol and dichloromethane were reported as the most abundant [40]. Methanolic extract from *Daucus carota*, obtained by sonication for 5 min, also yielded 4.7% [41]. Although a greater amount of *Daucus carota* biomass was used in that study, we observed that the methanolic extract of *P. dissectum* showed a higher yield, with a difference of 13.2%. However, in another study on *Notopterygium incisum*, the methanolic extract yielded 25.7% [42]. These results may depend on the type of extraction, as well as the source of the extract since callus cultures do not contain chlorophyll-like plants; on the other hand, in vitro cultures are a sustainable resource.

### 2.4. Total Phenolic Content and Antioxidant Activity

The determination of total phenolic content (TPC) and antioxidant activity by ABTS and DPPH was only carried out for methanolic extract of callus cultures (MCE). MCE had contents of 26.5 ± 0.4 mg GAE/g extract. Regarding antioxidant activity, it was shown that the OH^●^ elimination capacity was dependent on the concentration of the extract, i.e., there was a greater capacity to eliminate free radicals as the concentration of the extract increased. However, to exhibit the greatest radical elimination, the ABTS test required concentrations lower (15.6 µg/mL) than for DPPH (78.1 µg/mL) (Figure 4a,b). Regarding the mean maximum inhibitory concentration (IC_50_), the DPPH assay result was 49.4 ± 0.2 μg/mL, and the ABTS result was 10.0 ± 0.0 μg/mL. As expected, for both cases, the positive control (vitamin C) displayed lower IC_50_ values because it is a pure and potent antioxidant compound.

Antioxidant compounds are widely distributed in plants and have been studied for their positive health attributes; for example, it is known that adrenaline causes nitrative and oxidative stress by inducing damage to cellular genes, proteins, and lipids, resulting in damage to cardiomyocytes or a cardioprotective effect [43]. In addition, they improve enzymatic and non-enzymatic antioxidant systems, among others. On the other hand, it has also been reported that natural antioxidant compounds reduce oxidative stress and consequently increase insulin secretion and decrease DNA damage that can cause carcinoma [44]. As demonstrated in our study of *P. dissectum*, the extracts contain phenolics and flavonoids, which contribute to high antioxidant activity.

There are no studies in the literature reporting on TPC for *P. dissectum*; however, there are some related species of the Apiaceae family. For instance, a study on the methanolic extract of *Seseli rigidum*, using 10 g of fresh flower biomass, obtained a TPC of 76.62 mg GAE/g extract; furthermore, the assessment of its antioxidant activity using a DPPH assay resulted in an IC_50_ of 98.95 µg/mL [45]. Although the methanolic extract of *Seseli rigidum* had a higher TPC, the methanolic extract of *P. dissectum* exhibited better antioxidant activity, as shown in the DPPH test, and doubled its effectiveness in the inhibition of radical chromogens. In other species, such as *Capnophyllum peregrinum* L., the methanolic extract of the aerial part is reported to be abundant in TPC and to possess high antioxidant activity, as shown via DPPH and ABTS tests [46]. Therefore, it is observed that the MCE of *P. dissectum* exhibited better antioxidant activity than those of *S. rigidum* and *C. peregrinum* (L.), even with lower TPC. In contrast, methanolic extracts of wild and cultivated *Alepidea amatymbica* rhizomes also showed high TPC [47]. In another study, the methanolic leaf extract of *Eryngium foetidum* L. showed lower amounts of TPC with 16.09 mg GAE/g extract [48].

It is known that TPC depends on the strategy for each experiment, the species type, and the amount of biomass and solvent used, e.g., hydromethanolic extracts (MeOH-H_2_O) obtained from 30 g of aerial parts of *Petroselinum crispum*, *Apium graveolens*, and *Coriandrum sativum* had a TPC of 2.16 to 1.37 mg GAE/g extract. However, the extracts showed antioxidant activity, with IC_50_ values of 22.84 µg/mL (*P. crispum*) and 77.62 µg/mL (*C. sativum*), respectively, using the DPPH test [49]. Similarly, the hydromethanolic extract of *Ducrasia anethifolia* and the methanolic extract of *Bupleurum kaoi* demonstrated IC_50_ values of 15.22 µg/mL and 4.35 mg/mL, respectively, using the same method [50,51]. Regarding antioxidant activity, extracts of *P. crispum* and *D. anethifolia* presented better results than the MCE of *P. dissectum*, but the extract of *A. graveolens* showed similar effects to the MCE of *P. dissectum*, while *C. sativum* and *Bupleurum kaoi* extracts were less active. The Apiaceae family likely contains phenols attributable to antioxidant activity, including anti-inflammatory and antiproliferative properties.

### 2.5. Anti-Inflammatory Activity in RAW 264.7 Cells

#### 2.5.1. Effect of Extracts on Cell Viability

Cell viability is a prior step to the in vitro anti-inflammatory evaluation of the HCE, DCE, and MCE from the *P. dissectum* callus culture. None of the extracts at 5–40 µg/mL, including the control with 0.5% DMSO, affected cell viability (Figure 5a–c). In contrast, etoposide as a positive control decreased macrophage viability by 47.3 ± 5.2%.

However, it has been reported for *Cryptotaenia japonica* (Apiaceae) that using high concentrations of methanolic extract (400 µg/mL) causes cytotoxicity in macrophages [52]. This indicates that the MCE of *P. dissectum* could be cytotoxic at concentrations greater than 40 µg/mL. In contrast, the ethanolic extract of *Torilis japonica* did not have a significant cytotoxic effect on the viability of RAW 264.7 cells up to 75 µg/mL [53]. Similarly, the ethanolic extract of *Eryngium foetidum* leaves did not affect viability; in addition, the extract inhibited NO production from 35 µg/mL [54]. Despite the high concentrations of ethanolic extracts of *T. japonica*, *E. foetidum*, and *P. dissectum*, they did not affect the viability, suggesting that the ethanol, methanol, dichloromethane, and hexane extracts were safe at those concentrations.

#### 2.5.2. Effect of Extracts on the Inhibition of NO Production

To evaluate the inhibition of nitric oxide (NO) production, the extracts were evaluated at concentrations of 5 to 40 µg/mL. All extracts significantly decreased NO production compared to the control at 100% stimulation with LPS. A trend was observed in the inhibition of NO production as the concentration of the extracts increased; in fact, the HCE demonstrated greater inhibition of NO (28.6 ± 6.3%) at 40 µg/mL, followed by the DCE with 25 ± 8.4%, while indomethacin inhibited 49.7 ± 2.8% at 30 µg/mL (Figure 6a–c).

The present study represents the first report evaluating the inhibitory effect of NO in one of the species of the genus *Prionosciadium.* However, there have been studies undertaken on other species of the Apiaceae family, e.g., when the methanolic extract of *Anethum graveloens* (AGF) was evaluated at concentrations of 5–100 µg/mL, all concentrations decreased the NO concentration by values of 7.7–82.2% [55]. The methanolic root extract of *Angelicae dahuricae* decreased NO production by 45.1% when at concentrations of up to 200 μg/mL [56]. It is likely that the MCE of *P. dissectum* could have a higher percentage of NO inhibition using higher concentrations; however, these can also affect the cell viability, including the sample’s solubility. The ethanolic extracts of *Heracleum moellendorffii* and *Torilis japonica* have also demonstrated a decrease in NO levels in RAW 264.7 cells activated with LPS. The *H. moellendorffii* extract evaluated between 12.5 and 50 µg/mL demonstrated NO inhibition between 20.3 and 88.4%, while the *T. japonica* extract at 75 µg/mL presented 90% NO inhibition [53,57]. The above-mentioned studies report a variety in the concentrations of the extracts evaluated, with some concentrations used in those studies being higher than those evaluated for *P. dissectum*, which indicates that, in our study, we used relatively low concentrations of extracts (less than 50 µg/mL) and still obtained favorable responses in terms of anti-inflammatory activity in vitro.

### 2.6. Effect of Extracts on Antiproliferative Cancer Cells

*P. dissectum* callus extracts were evaluated at concentrations of 0.01–100 µg/mL; however, after 48 h of treatment, a significant effect was only found at 100 µg/mL (Table 1). Dichloromethane extract (DCE) stood out as exhibiting the greatest antiproliferative effect on cancer cell lines, specifically for Hep3B, CasKi, and HeLa, inhibiting 24.3 ± 4.0, 19 ± 3.6, and 18.7 ± 2.9%, respectively, while, for the methanolic extract (MCE), its best level of inhibition was shown for A549 (19 ± 3.0%) and Hep3B (19.0 ± 3.0%). Hexane extract (HCE) also had a greater antiproliferative effect on Hep3B, with 21.7 ± 2.9%. The HCE showed the most significant inhibition (16.0 ± 1.0%) on the HepG2 cell line. Similarly, in IHH (non-cancerous cells), the extracts presented 15.3–16.7% antiproliferative effects, indicating that those extracts have low selectivity.

Previous studies on the Apiaceae family have reported their antiproliferative effect on cancer cell lines. The acetone root extract of *Bupleurum scorzonerifolium* showed an antiproliferative effect in A549, with IC_50_ = 59 µg/mL [58]. In another study, acetone extract from the roots of *B. scorzonerifolium* showed antiproliferative activity and apoptosis in A549, with an IC_50_ value of 60 µg/mL [59]. Saikosaponin D isolated from the *Bupleurum* genus has also been reported to present 73.9% inhibition in the proliferation of A549 cells at 20 µM at 48 h, and its maximum effect was at 72 h with 83.7% inhibition [60]. In this way, we can notice that extracts and/or pure compounds of this type of species present favorable antiproliferative results, although this will depend on the type of extract and the part of the plant. Petroleum ether and chloroform fractions derived from the ethanolic extract of *Anthriscus sylvestris* roots showed IC_50_ values in the range of 18.25–45.66 µg/mL in HepG2 and HeLa cells [61]. The methanolic extracts of *Centella asiatica* leaf showed an antiproliferative effect in HeLa and A549, with IC_50_ values of 78.30 and 68.51 µg/mL, respectively [62]. In contrast, the methanolic extract of *Coriandrum sativum* L. leaves did not inhibit the proliferation of HepG2 cells at 10 µg/mL [63]. However, the methanolic extract of *C. asiatica* L. demonstrated greater inhibition in HeLa and A549 cancer cells than the DCE and MCE of *P. dissectum*; moreover, HCE, MCE, and DCE showed an inhibitory effect in the HepG2 cell line at 100 µg/mL.

In this pioneering study, we successfully established a callus culture of *Prionosciadium dissectum*, representing the first report of in vitro culture for both the species and the *Prionosciadium* genus. Furthermore, we demonstrated that the extracts may be able to scavenge free radicals, as well as inhibit inflammation and cancer cell proliferation at certain extract concentrations. However, more targeted studies are needed to characterize the chemical compounds in the extracts and determine which compounds are involved in these biological activities. This first study lays the groundwork for future studies of this important genus.

## 3. Materials and Methods

### 3.1. Plant Material

Whole plants, including roots, flowers, and leaves, were collected in August 2018 in the mountains of the municipality of Sayula, Jalisco, Mexico. The geographic coordinates are 19°49′31″ LN and 103°37′57″ LW at 1890 m.a.s.l. (Figure 7).

Specimen identification was carried out by Dr. Daniel Sánchez Carbajal and Dr. Pablo Carrillo Reyes of the Laboratorio Nacional de Identificación y Caracterización Vegetal (Laniveg-Conacyt) and the “Luz María Villareal de Puga” Herbarium (IBUG), Instituto de Botánica, Universidad de Guadalajara. The plant was identified as *Prionosciadium dissectum* J.M. Coult. & Rose (Apiaceae), and a sample was deposited in the herbarium with the registration number 209558. Another sample was collected and immediately transplanted into a pot containing substrate from the collection site and was maintained outdoors in the back garden (Figure 1a). The plant grew successfully; this was then used as a resource for leaf explants for callus induction experiments.

### 3.2. Callus Induction

The culture medium consisted of Murashige and Skoog (MS) [65], and 3% sucrose (Sigma-Aldrich, Inc. St. Louis, MO, USA) was added. To induce calluses, different concentrations of plant growth regulators (PGRs) were evaluated on leaf explants, using 2,4-dichlorophenoxyacetic acid (2,4-D) as an auxin and kinetin (KIN) as a cytokinin (Sigma-Aldrich, Inc. St. Louis, MO, USA). Concentrations of 0.5, 1.0, and 1.5 mg/L of auxin and cytokinin were tested individually and in combination, including a control (PGRs-free). The MS medium was adjusted to a pH of 5.8, gelled with 2 g/L Phytagel^®^ (Sigma-Aldrich, Inc. St. Louis, MO, USA), and then transferred to Gerber jars, each containing 25 mL MS medium. The culture medium was sterilized in an autoclave at 120 °C and 18 psi for 18 min. Juvenile leaves of *P. dissectum* were sectioned into small portions and washed with neutral detergent Hyclin-Plus (Hycel^®^; Guadalajara, Jalisco, Mexico) for 5 min, followed by immersion of 10% (*v*/*v*) NaClO (Cloralex^®^) for 15 min. Finally, they were rinsed with sterile distilled water in a horizontal laminar flow cabinet. Leaf portions were sectioned into sizes of approximately 15 mm in length, and 4 explants were placed into each Gerber jar containing 25 mL of gelled sterile culture medium. Each treatment consisted of 4 jars (16 explants), and the experiment was repeated twice. All cultures were incubated in a 16 h light photoperiod at 25 ± 2 °C.

### 3.3. Callus Culture Establishment and Growth Kinetics

The best callus induction treatment, i.e., calluses showing better growth and a friable appearance, were frequently subcultured every 20 or 30 days for six months, using the culture medium with the same induction treatment. Subsequently, a growth kinetics analysis was carried out to identify the different growth phases. The 30-day-old calluses were sown in several Gerber-type flasks, inoculated with 1.5 g of callus, and incubated for 45 days. Four flasks were harvested every five days, the biomass was washed three times using distilled water and filter paper, and the excess water was removed using a vacuum pump. The biomass was dried for 24 h in an oven at 40 °C, and the dry weight data were recorded. The experiment was repeated twice, and the dry weight data were used to plot the growth curve.

### 3.4. Callus Biomass Preparation and Extraction Process

After establishing the growth kinetics, frequent subcultures were performed, and calluses were harvested every 30 days for maximum biomass production. Callus biomass was washed and dried as described in Section 2.3 until we obtained at least 50 g of dry biomass. For extraction, the dry callus biomass (50 g) was ground to a fine powder in a pestle and mortar and transferred to a 500 mL Erlenmeyer flask. Successive extractions were carried out by maceration with hexane, dichloromethane, and methanol, respectively; each extraction 48 h, and the extraction was carried out twice on the same sample. All reagents were reagent-grade, with a purity greater than 99.8% (Fermont^®^, Monterrey, Nuevo Leon, Mexico). The biomass/volume ratio was 1 g dry biomass per 10 mL solvent (1:10). The corresponding extracts were mixed, filtered through filter paper, and concentrated under reduced pressure in a Rotavapor (Büchi EL 131; BÜCHI Labortechnik AG, Flawil, CH, Switzerland) and Water Bath (Büchi 461). Finally, three extracts were obtained: hexane callus extract (HCE), dichloromethane callus extract (DCE), and methanolic callus extract (MCE). All extracts were dried at 40 °C until they reached a constant weight.

### 3.5. Determination of Total Phenolic Content

Total phenol content (TPC) was only evaluated for MCE. TPC was determined using the Folin–Ciocalteu (FC) (Sigma-Aldrich, Inc., St. Louis, MO, USA) reagent [66]. A sample of MCE was dissolved in methanol (2.5 mg/mL); then, a 100 µL aliquot was mixed with 200 µL FC (2 N) and 2.0 mL distilled H_2_O. The mixture was incubated at 25 °C for 3 min in dark in an AGO-6040 (SEV-PRENDO™, Puebla, Mexico) orbital shaker at 115 ± 1 rpm. Next, 1.0 mL Na_2_CO_3_ (20%; *w*/*v*) was added and incubated again under the conditions already mentioned. Samples were immediately read against a blank on a DR-5000™ UV-Vis spectrophotometer (HACH^®^, London, ON, Canada) at 765 nm. Quantification was performed using a calibration curve of gallic acid (GA) in ranges of 6.25 to 500 mg/L (R^2^ = 0.9997). The results are expressed as mg equivalents of gallic acid (GAE) per gram of dry extract (mg GAE/g extract).

### 3.6. In Vitro Assessment of Antioxidant Activity

The determination of the antioxidant activity of methanolic callus extract (MCE) was carried out using 2,2-diphenyl-1-picrylhydrazyl (DPPH) and 2,2′-azino-bis(3-ethylbenzothiazoline-6-sulfonic acid) (ABTS·^+^) purchased from Sigma-Aldrich, Inc., St. Louis, MO, USA [67]. A 1650 µL aliquot of MCE (4.9–78.1 µg/mL) or positive control (vitamin C: 1.6–6.3 µg/mL) was dissolved in methanol and then mixed with 1650 µL DPPH (0.1 mM). The mixture was shaken vigorously for 1 min in a vortex and incubated in the dark at 25 °C for 30 min on an AGO-6040 (Prendo™; Puebla, Mexico) orbital shaker at 115 ± 1 rpm. Immediately, samples were read at 517 nm. The results are expressed as the mean inhibitory concentration (IC_50_) of extracts.

For the ABTS method, a stock solution was prepared by mixing an ABTS solution (7 mM) with potassium persulfate (K_2_S_2_O_8_: 2.45 mM) in a 1:1 ratio. The mixture was left stirring at 115 ± 1 rpm, in the dark at 25 °C for 12 to 16 h. The ABTS·^+^ working solution was obtained by diluting ABTS·^+^ stock solution with phosphate-buffered saline (PBS, pH 7.4), adjusting the absorbance to 0.7 ± 0.02 at 734 nm. Subsequently, a 330 μL aliquot of MCE (3.9–15.6 μg/mL) or vitamin C (1.3–4 μg/mL) was mixed with 2.97 mL ABTS·^+^ working solution and shaken vigorously. The mixture was incubated for 6 min under the same conditions as above, and then samples were read at 734 nm. The results were expressed as the mean inhibitory concentration (IC_50_) of extracts.

### 3.7. In Vitro Anti-Inflammatory Activity

All extracts (HCE, DCE, and MCE) were used for anti-inflammatory and antiproliferative studies. RAW 264.7 murine macrophage cells (Tib-71™, ATCC^®^, Manassas, VA, USA) were used. Cells were cultured in DMEM/F12 medium supplemented with 10% heat-inactivated fetal bovine serum (FBS) and antibiotic-free. Cells were maintained in a humid atmosphere with 5% CO_2_ at 37 °C and subcultured in Corning^®^ 25 cm^2^ flasks.

Cells were cultured in 96-well microplates (10,000 cells/well) containing 0.1 mL culture medium and incubated for 24 h. Next, cells were treated with the extracts (5–40 µg/mL), DMSO (0.21%, *v*/*v*) as a vehicle, or etoposide (40 µg/mL) as a positive control and then incubated for 22 h. Cell viability was determined using the 3-(4,5-dimethylthiazol-2-yl)-5-(3-carboxymethoxyphenyl)-2-(4-sulfophenyl)-2H-tetrazolium (MTS) assay from PROMEGA^®^ (Thermo Fisher Scientific Inc, Waltham, MA, USA). Briefly, 20 µL of MTS solution was added to each well and incubated for 2 h. Optical density was measured at 490 nm in an ELISA Microplate Reader.

Cells were cultured in a 96-well microplate (20,000 cells/well) containing 0.2 mL culture medium and incubated for 24 h. Then, cells were treated with the extracts (without affecting the viability), DMSO (0.21%, *v*/*v*), or indomethacin (INDO) (30 µg/mL), both of which were purchased from Sigma-Aldrich, Inc. (St. Louis, MO, USA), and incubated for 1 h. Next, the pro-inflammatory stimulus with lipopolysaccharide (LPS, 4 µg/mL) was added into wells with treatments (extracts, DMSO, or INDO). Cells treated only with LPS were considered to be 100% stimulation and untreated cells were considered to be negative controls, and both were incubated at 37 °C for 20 h. Finally, cell-free supernatants were collected and used as fresh to quantify nitric oxide (NO).

Nitrite, a stable product of nitric oxide (NO), was used as an indicator of NO production in cell supernatants and was measured according to Griess reagent (Invitrogen, Thermo Fisher Scientific, Inc., Waltham, MA, USA) [68]. Briefly, in a 96-well microplate, 50 µL of each supernatant was mixed with 100 µL of Griess reagent, i.e., 50 µL sulfanilamide (1%) and 50 µL N-(1-Naphthyl) ethylenediamine dihydrochloride (0.1%) in phosphoric acid solution (2.5%), and incubated for 10 min at room temperature. The optical density was measured at 540 nm (OD_540_) in an ELISA microplate reader. Nitrite concentration was calculated by comparison with OD_540_ using a calibration curve of NaNO_2_ prepared with fresh culture medium.

### 3.8. Antiproliferative Activity on Cancer Cells

*P. dissectum* callus extracts were evaluated against the Hep3B and HepG2 (hepatocellular), CasKi (human papillomavirus type 16: HPV-16), A549 (lung), and HeLa (cervical) human cancer cell lines, which were obtained from ATCC (American Type Culture Collection, Manassas, VA, USA). We also included an immortalized human hepatocyte (IHH) cell line as a control, representing noncancerous cells [69]. Hep3B, HepG2, and IHH cells were grown in minimum essential medium (Invitrogen, Thermo Fisher Scientific, Inc., Waltham, MA, USA), A549 cells were grown in DMEM/F12 medium (Invitrogen, Thermo Fisher Scientific, Inc., Waltham, MA, USA), and HeLa cells were grown in Dulbecco’s modified high-glucose Eagle’s medium (DMEM HG, Caisson Labs, Smithfield, UT, USA). All culture media were supplemented with 10% (*v*/*v*) fetal bovine serum (Biowest LLC, Riverside, MO, USA) and 2 mM glutamine. All the cultures were incubated at 37 °C in a humid atmosphere with 5% CO_2_. Cells, initially 8000 cells per well in a 96-well plate, were incubated for 24 h. Next, cells were exposed for 48 h in the presence or absence (negative control) of callus extracts solubilized in DMSO (0.01%) at concentrations of 0.01, 0.1, 1.0, 10, or 100 µg/mL. Paclitaxel (Sigma Aldrich, St. Louis, MO, USA) was used as the positive control. When determining the number of viable cells in proliferation, we used a [3-(4,5-dimethylthiazol-2-yl)-5-(3-carboxymethoxyphenyl)-2-(4-sulfophenyl)-2H-tetrazolium] inner salt MTS assay kit (Promega, Madison, WI, USA) and followed the manufacturer’s instructions. Cell viability was determined by absorbance at 450 nm using an automatic microplate reader (Promega, Madison, WI, USA). The experiments were conducted in triplicate with three independent experiments.

### 3.9. Statistical Analysis

An analysis of variance was performed on all callus induction data, followed by a Tukey test using the SAS software, version 9.0. Dunnett’s test was performed on the antioxidant and anti-inflammatory activity data using the statistical software Prism version 8.01, and the IC_50_ values were determined by regression analysis using the same software.

## 4. Conclusions

This study successfully established a callus culture of *P. dissectum*, obtaining friable callus with 2,4-D and kinetin. The methanolic extract of this callus demonstrated notable phenolic content and significant antioxidant activity, comparable to vitamin C. Furthermore, while non-toxic to macrophage cells, the extracts, particularly the hexane extract, effectively reduced nitric oxide production, suggesting their anti-inflammatory potential. Notably, all extracts exhibited significant antiproliferative effects against several cancer cell lines. However, further research, especially on methanolic extracts from both callus and wild plants, is warranted to confirm their impact on cell viability and toxicity. Overall, these preliminary findings suggest that *P. dissectum* callus cultures could be a valuable source of novel bioactive compounds with potential anti-inflammatory and anticancer properties.

## Figures and Tables

**Figure 1 plants-14-01394-f001:**
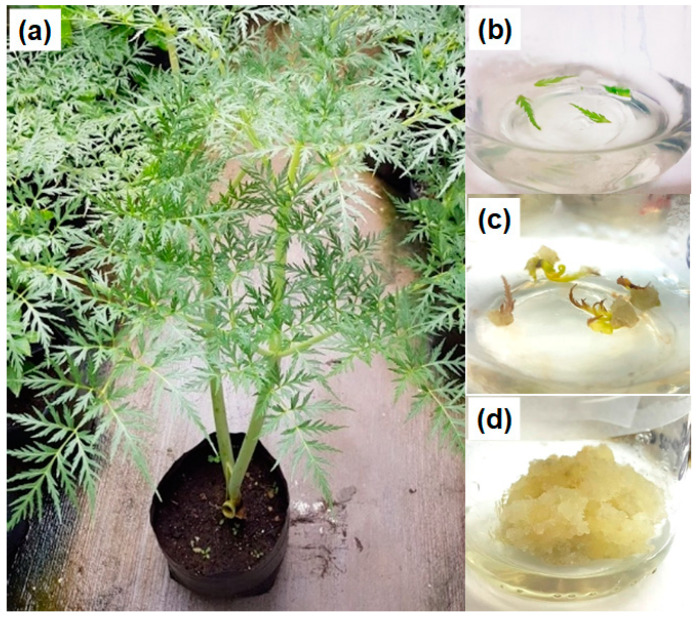
Plant and in vitro culture of *Prionosciadium dissectum*. Wild plant grown in a pot (**a**); leaf explants during the first day of culture in MS medium (**b**); leaf explant treated with 1 mg/L 2,4-D and 1 mg/L KIN exhibiting callus formation after 15 days of culture (**c**); friable callus development after 30 days of culture (**d**).

**Figure 2 plants-14-01394-f002:**
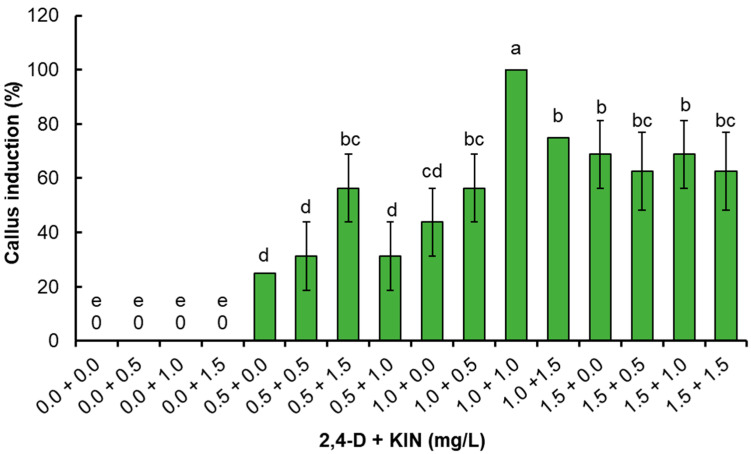
Effect of 2,4-D and KIN on the percentage of callus induction in leaf explants of *P. dissectum*. Tukey’s honest significance test. Columns with the same letter are not significantly different (*p* ≤ 0.05).

**Figure 3 plants-14-01394-f003:**
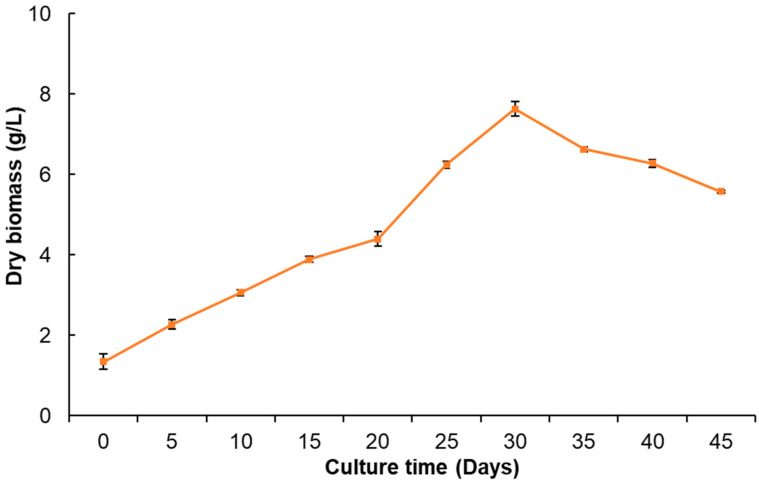
Growth kinetics of *P. dissectum* callus culture for 45 days in MS medium with 1.0 mg/L 2,4-D and 1.0 mg/L KIN.

**Figure 4 plants-14-01394-f004:**
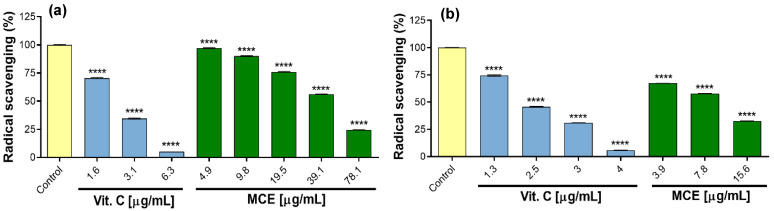
Hydroxyl radical scavenging activity of methanol callus extract (MCE). (**a**) DPPH test; (**b**) ABTS test. The significant difference was determined using an ANOVA followed by a Dunnett’s test. Vitamin C (Vit. C) and MCE compared with the control (**** *p* ˂ 0.0001).

**Figure 5 plants-14-01394-f005:**
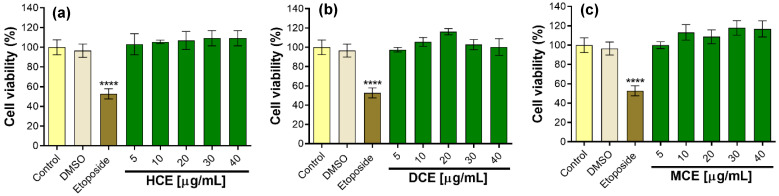
Effect of callus extracts on cell viability of RAW 264.7 macrophages. Values represent mean ± standard deviation of three independent experiments (*n* = 3). The significant difference was determined using an ANOVA followed by a Dunnett’s test. DMSO, ETOP (etoposide), and extracts compared with the control group (**** *p* < 0.0001). Control = cells without treatment, defined as 100% viability. (**a**) HCE = hexane callus extract; (**b**) DCE = dichloromethane callus extract; (**c**) MCE = methanol callus extract.

**Figure 6 plants-14-01394-f006:**
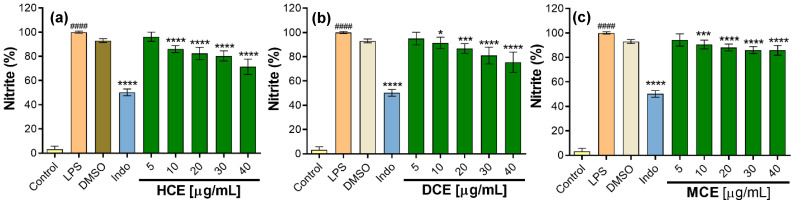
Effect of callus extracts on nitric oxide (NO) production in RAW 264.7 macrophages stimulated with LPS. Values represent the mean ± SD of three independent experiments (*n* = 3). The significant difference was determined using an ANOVA followed by a Dunnett’s test. LPS compared to the control group (#### *p* < 0.0001); DMSO (dimethylsulfoxide), INDO (indomethacin), and extracts compared with the LPS group (* *p* < 0.05, *** *p* < 0.001, and **** *p* < 0.0001). Control = cells without stimulation. (**a**) HCE = hexane callus extract, (**b**) DCE = dichloromethane callus extract, (**c**) MCE = methanolic callus extract.

**Figure 7 plants-14-01394-f007:**
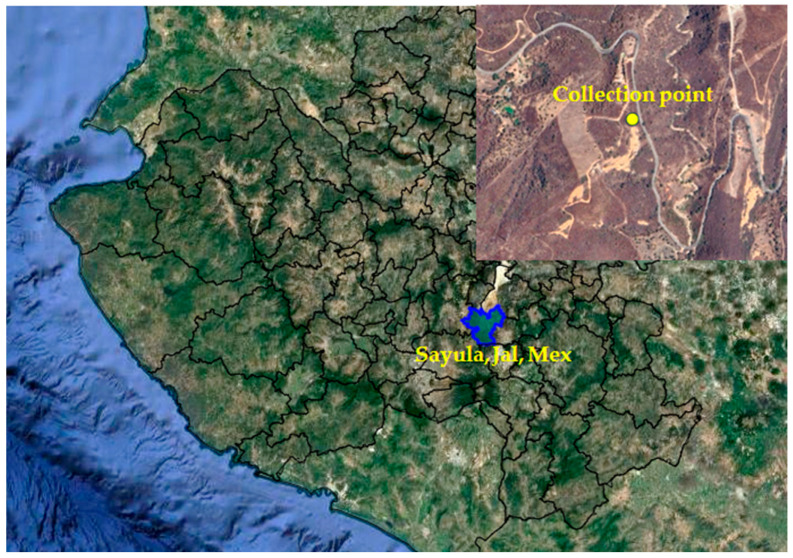
Satellite image showing the geographic area where *Prionosciadium dissectum* was collected. Prepared by the authors with images from INEGI [64].

**Table 1 plants-14-01394-t001:** Antiproliferative effect of *P. dissectum* callus extracts on cancer cell lines evaluated at 100 µg/mL.

Extract	Inhibition (%)					
	IHH	Hep3B	HepG2	CasKi	HeLa	A549
HCE	15.3 ± 1.5	21.7 ± 2.9	16.0 ± 1.0	17.3 ± 2.5	17.3 ± 3.5	18.3 ± 1.5
DCE	15.3 ± 1.5	24.3 ± 4.0	14.0 ± 3.6	19.0 ± 3.6	18.7 ± 2.9	17.7 ± 3.1
MCE	16.7 ± 2.9	19.0 ± 1.7	15.0 ± 3.5	18.0 ± 1.0	17.0 ± 3.0	19.0 ± 3.0

HCE = hexanic callus extract; DCE = dichloromethanic callus extract; MCE = methanolic callus extract; IHH = immortalized human hepatocytes; Hep3B and HepG2: hepatocellular, CasKi (HPV-16), HeLa (cervical), A549 (lung).

## Data Availability

Data are contained within the article.

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
