# Peer review of "Antioxidant, Anti-Inflammatory, and Antiproliferative Activity of a Callus Culture of *Prionosciadium dissectum* (Apiaceae)"

_plants, 2025, doi:10.3390/plants14091394_

Round 1
Reviewer 1 Report
Comments and Suggestions for Authors
The article "Antioxidant, Anti-inflammatory, and Antiproliferative Activity of a Callus Culture of Prionosciadium dissectum (Apiaceae)" has interesting results that warrant publication in the journal. The introduction is appropriate for the topic, and the methodology is well-defined with the stated objectives. However, I have some concerns:
1. Why did the authors specifically choose to evaluate these extract concentrations and not others?
2. I suggest the authors discuss the results based on the active ingredients reported in the literature with the pharmacological effects found with these extracts. This may have a pharmacodynamic and/or pharmacokinetic explanation.
3. The conclusion seems more like a summary of the results; please be more specific.
4. Add doi to the references.
Author Response
Response to Reviewer 1 Comments
General comment.
The article "Antioxidant, Anti-inflammatory, and Antiproliferative Activity of a Callus Culture of Prionosciadium dissectum (Apiaceae)" has interesting results that warrant publication in the journal. The introduction is appropriate for the topic, and the methodology is well-defined with the stated objectives. However, I have some concerns:
Comment 1. Why did the authors specifically choose to evaluate these extract concentrations and not others?
Response 1. To evaluate the antiproliferative activity, the extracts were first screened using concentrations of 0.01, 0.1, 1.0, 10, and 100 µg/mL, but only the 100 mg/mL concentration was the most promising, so all extracts were only evaluated at that concentration in all cancer cell lines. This is specified in the methodology and results section. Lines 277-278 y 455-457.
Comment 2. I suggest the authors discuss the results based on the active ingredients reported in the literature with the pharmacological effects found with these extracts. This may have a pharmacodynamic and/or pharmacokinetic explanation.
Response 2. This study aimed to induce a callus culture of P. dissectum, obtain extracts, and evaluate their antioxidant, antiproliferative, and anti-inflammatory activities. Given the current lack of data on the chemical composition of such callus extracts, a discussion of pharmacokinetic or pharmacodynamic effects is premature for this initial investigation. However, a future phytochemical analysis will be conducted, which will then allow for a pharmacokinetic evaluation and appropriate discussion. Nevertheless, this was not within the objectives of this initial study.
Comment 3. The conclusion seems more like a summary of the results; please be more specific.
Response 3. The section conclusion was rewritten.
Comment 4. 4. Add doi to the references.
Response 4. All references were reviewed, and the DOI was added to each reference where appropriate. The DOI are highlighted in yellow in the References list

Reviewer 2 Report
Comments and Suggestions for Authors
The paper entitled “Antioxidant, Anti-inflammatory and Antiproliferative Activity of a Callus Culture of Prionosciadium dissectum (Apiaceae)” describes an induction of callus from P. dissectum leaves, evaluation of the antioxidant, anti-inflammatory, and antiproliferative activity of obtained extracts. The paper is carefully written, actual, presents the new data on extraction and evaluation of biologically potent properties of a Callu s Culture of Prionosciadium dissectum (Apiaceae). The paper could be accepted to Plants after revision. The topic of this manuscript is original and possess novelty since the search of new the extracts with antiproliferative activity is important. The main question addressed by this research is the search of new biomass with antiproliferative and other types of activities. The conclusions consistent with the evidence and are detailed, the necessary arguments are presented and addressed to the main question posed. The figures and tables are concisely and clearly reflect the details of the discussed topic. The recommendations to the authors are the following: 1) the definitions of MS and BA should be provided where its 1st mentioned (Line 106 and 277); 2) Line 109 - give an explanation, or rephrase of “were combined at concentrations of 0.0”3) Line 262 in the description of figure 1, “after 15 days of culture (b)” should be “after 15 days of culture (c)”; 4) Figure 2 is missed. Please correct the numbering of figures. 5) The results and discussion section should contain the results of your own research, rather than a large amount of information data. Sections 3.1 – 3.6 contain a lot of unnecessary information for other genera of the Apiaceae family, which are not related to this study, and only confuse the reader with a large amount of number data. There is no clear relationship of which P. dissectum is compared to other types. Each section contains information about various representatives that is not, for example, mentioned in the following section. It is unclear why all this data is presented if it is not discussed or compared with the results of the current study.6) Section 3.6: the authors claim that “the hexane extract had the greatest effect on HepG2 with 16 ± 1%.” However, as can be seen in table 1, the best % of inhibitory was observed for Hep3B with 21.7%. Please clarify.
Author Response
|
Response to Reviewer 2 Comments
|
General comment. The paper entitled “Antioxidant, Anti-inflammatory and Antiproliferative Activity of a Callus Culture of Prionosciadium dissectum (Apiaceae)” describes an induction of callus from P. dissectum leaves, evaluation of the antioxidant, anti-inflammatory, and antiproliferative activity of obtained extracts. The paper is carefully written, actual, presents the new data on extraction and evaluation of biologically potent properties of a Callu s Culture of Prionosciadium dissectum (Apiaceae). The paper could be accepted to Plants after revision. The topic of this manuscript is original and possess novelty since the search of new the extracts with antiproliferative activity is important. The main question addressed by this research is the search of new biomass with antiproliferative and other types of activities. The conclusions consistent with the evidence and are detailed, the necessary arguments are presented and addressed to the main question posed. The figures and tables are concisely and clearly reflect the details of the discussed topic. The recommendations to the authors are the following:
Comment 1. 1) the definitions of MS and BA should be provided where its 1st mentioned (Line 106 and 277);
Response 1. The corresponding acronyms MS and BA were added to the text. Lines 336 and 118-119.
Comment 2. 2) Line 109 - give an explanation, or rephrase of “were combined at concentrations of 0.0”
Response 2. The paragraph was rephrased. Lines 340-341.
Comment 3. 3) Line 262 in the description of figure 1, “after 15 days of culture (b)” should be “after 15 days of culture (c)”;
Response 3. Thank you. This was corrected in Figure 1. Lines 102-104.
Comment 4. 4) Figure 2 is missed. Please correct the numbering of figures.
Response 4. Thanks for your observation. All figures have been revised. Because the Materials and Methods section was moved after the Results and Discussion section, the order of the figures changed, and Figure 7 was added to the manuscript.
Comment 5. 5) The results and discussion section should contain the results of your own research, rather than a large amount of information data. Sections 3.1 – 3.6 contain a lot of unnecessary information for other genera of the Apiaceae family, which are not related to this study, and only confuse the reader with a large amount of number data. There is no clear relationship with which P. dissectum is compared to other types. Each section contains information about various representatives that is not, for example, mentioned in the following section. It is unclear why all this data is presented if it is not discussed or compared with the results of the current study.
Response 5. We agree with the comment. Sections 3.1-3.6 (currently Sections 2.1-2.6) contain information not directly applicable to P. dissectum, the focus of this study. However, as clarified in lines 217-222, 193-196, 259-261, due to the lack of studies for the genus Prionosciadium, we initially included research on closely related species within the Apiaceae family, and in very specific cases to other species. Moreover, to prevent confusion, some irrelevant numerical data from other species were subsequently removed. Specifically for the discussion of the anti-inflammatory and antiproliferative activity section, we believe it is important to retain the data. Please accept this information in the manuscript. All modifications are highlighted in yellow in each section. Lines 120-137, 144-148, 163-165, 217-222, 241-244, 271-275, 277-284, 306-309.
Comment 6. 6) Section 3.6: the authors claim that “the hexane extract had the greatest effect on HepG2 with 16 ± 1%.” However, as can be seen in table 1, the best % of inhibitory was observed for Hep3B with 21.7%. Please clarify.
Response 6. All texts have been paraphrased for clarity. Currently it is Section 2.6. Lines 277-284.

Reviewer 3 Report
Comments and Suggestions for Authors
The paper is focused on Antioxidant, Anti-inflammatory and Antiproliferative Activity of a Callus Culture of Prionosciadium dissectum (Apiaceae). Few of my main concerns are listed below.
- iThenticate similarities of 31% is much too high for an original paper. Revise the entire manuscrupt in this regard.
- The fact “that humanity has used since ancient times” the plants in traditional medicine must be referenced. I suggest checking and referring to Bungau, S.G.; Popa, V.-C. Between Religion and Science: Some Aspects Concerning Illness and Healing in Antiquity. Transylvanian Review, 2015, 26(3), 3-19. (accessible https://www.researchgate.net/publication/286442576_Between_Religion_and_Science_Some_Aspects_Concerning_Illness_and_Healing_in_Antiquity.).
- A satellite image of the geographic area would be relevant would be relevant.
- Section 2. should be completed as Materials and apparatus. Also, please provide complete information regarding the apparatus and chemicals used (maybe in a separate subsection) in the experimental stage:
- the Model, Producer/manufacturer, City, and Country for EACH/ALL APPARATUS (4 information) used in the research and
- the Producer, Country, purity degree, and concentration or CAS (4 information) for EACH REAGENT/chemical used in the research (if the case). Check the entire manuscript in this regard. This information gives the possibility for replicating you experiment to other authors and are requested in ALL journals.
- It is not needed subdividing section 2.7. in subsections of few lines. Merge them.
- Subsection 2.8. All the statistic softs and the other programs/their variants used for analysis must be provided and referenced.
- Discussion part is very poor vs. Results. I suggest the following:
- Compare your data with similar already published
- Considering the bioactive compounds in Prionosciadium dissectum you have focused on, detail the biomedical role of the antioxidant compounds using Abdel-Daim et al. both papers https://doi.org/10.1155/2018/2098123 and https://doi.org/10.1155/2018/6276438
- After L 456, as a last paragraph of Discussion, a paragraph detailing the strengths and limitations of your research must be added.
Author Response
Response to Reviewer 3 Comments
Comment 1. The paper is focused on Antioxidant, Anti-inflammatory and Antiproliferative Activity of a Callus Culture of Prionosciadium dissectum (Apiaceae). Few of my main concerns are listed below.
Thenticate similarities of 31% is much too high for an original paper. Revise the entire manuscrupt in this regard.
Response 1. This is probable; however, some plagiarism detection software may incorrectly identify reference lists as plagiarism despite the inclusion of corresponding citations within the text. However, the entire text was reviewed and paraphrased by a native English speaker for clarity. A certificate is attached.
Comment 2. The fact “that humanity has used since ancient times” the plants in traditional medicine must be referenced. I suggest checking and referring to Bungau, S.G.; Popa, V.-C. Between Religion and Science: Some Aspects Concerning Illness and Healing in Antiquity. Transylvanian Review, 2015, 26(3), 3-19. (accessible https://www.researchgate.net/publication/286442576_Between_Religion_and_Science_Some_Aspects_Concerning_Illness_and_Healing_in_Antiquity.).
Response 2. We reviewed the suggested document and subsequently paraphrased the text (Lines 24-25), based on our understanding that citations are generally not appropriate within an abstract. However, this document will be considered in future reports. Thank you for the suggestion.
Comment 3. A satellite image of the geographic area would be relevant would be relevant.
Response 3. This was done. A satellite image of the geographic collection area was added as Figure 7, pag. 9.
Comment 4. Section 2. should be completed as Materials and apparatus. Also, please provide complete information regarding the apparatus and chemicals used (maybe in a separate subsection) in the experimental stage: the Model, Producer/manufacturer, City, and Country for EACH/ALL APPARATUS (4 information) used in the research and the Producer, Country, purity degree, and concentration or CAS (4 information) for EACH REAGENT/chemical used in the research (if the case). Check the entire manuscript in this regard. This information gives the possibility for replicating you experiment to other authors and are requested in ALL journals.
Response 4. Thank you for the suggestion. We have reviewed the comment; however, we believe it would be more appropriate to add this information to each section where appropriate. That is, the requested information is described in each section of the manuscript where the necessary equipment or other materials were used. Furthermore, all chemicals used in this research are reagent grade with a purity greater than 99.5%. All materials, chemicals, and equipment used are highlighted in yellow in each section, including the manufacturer and country.
Comment 5. It is not needed subdividing section 2.7. in subsections of few lines. Merge them.
Response 5. This was done. The subsections were merged and the numbering was corrected. Currently it is 3.7 instead of 2.7. Line 410.
Comment 6. Subsection 2.8. All the statistic softs and the other programs/their variants used for analysis must be provided and referenced.
Response 6. For statistical analysis, only the SAS and Prisma statistical programs mentioned in the section were used. No other software or variants were used.
Comment 7. Discussion part is very poor vs. Results. I suggest the following:
Compare your data with similar already published
Response 7. We have revised the manuscript and expanded the discussion. However, a detailed comparison with previous studies is not feasible, as this is the first report on callus culture and extract evaluation of P. dissectum. Moreover, no callus culture studies have been reported for other species within the same genus, making direct comparisons inappropriate. Nonetheless, we have incorporated references to studies involving extracts from related species. It is important to note that these studies are based on wild plants rather than callus cultures, and thus only allow for a limited comparative discussion. Lines 134-137, 158-160, 206-207, 241-244, 259-261, 294-298.
Comment 8. Considering the bioactive compounds in Prionosciadium dissectum you have focused on, detail the biomedical role of the antioxidant compounds using Abdel-Daim et al. both papers https://doi.org/10.1155/2018/2098123 and https://doi.org/10.1155/2018/6276438.
Response 8. Thank you for your comment. However, because the objective of this study was not to characterize the chemical composition of P. dissectum, we currently do not know the possible compounds that may be acting as free radical scavengers. Only total phenolics and flavonoids were determined using colorimetric methods, but we will report the detailed chemical composition in future studies. However, we have added a brief discussion with suggested references. Lines 186-194.
Comment 9. After L 456, as a last paragraph of Discussion, a paragraph detailing the strengths and limitations of your research must be added.
Response 9. This was done. A paragraph with the strengths and limitations was added. Lines 310-317

Round 2
Reviewer 1 Report
Comments and Suggestions for Authors
The authors responded to the suggestions.
Reviewer 3 Report
Comments and Suggestions for Authors
The authors improved their manuscript